# MedSegDiff: Medical Image Segmentation with Diffusion Probabilistic Model

**Junde Wu**[1,2,3]                                               JUNDEWU@IEEE.ORG
**Rao Fu**[1,2]                                                  1098712828@QQ.COM
**Huihui Fang**[1,2]                                          FANGHUIHUIBIT@163.COM
**Yu Zhang**[4]                                              ZHANGYUHIT2@HIT.EDU.CN
**Yehui Yang**[3]                                             YANGYEHUI@BAIDU.COM
**Haoyi Xiong**[3]                                           XIONGHAOYI@BAIDU.COM
**Huiying Liu**[5]                                        LIUHY@I2R.A-STAR.EDU.SG
**Yanwu Xu**[*1,2,6]                                              YWXU@IEEE.ORG

[1] *School of Future Technology, South China University of Technology, Guangzhou, China*

[2] *Pazhou Lab, Guangzhou, China*

[3] *Baidu Inc., Beijing, China*

[4] *Department of Electronic Science and Technology, Harbin Institute of Technology, Harbin, China*

[5] *Institute for Infocomm Research, A*STAR, Singapore*

[6] *Singapore Eye Research Institute, Singapore*

**Editors:** Accepted for publication at MIDL 2023

## Abstract

Diffusion Probabilistic Model (DPM) has recently become one of the hottest topics in computer vision. Its image generation applications, such as Imagen, Latent Diffusion Models, and Stable Diffusion, have demonstrated impressive generation capabilities, which have sparked extensive discussions in the community. Furthermore, many recent studies have found DPM to be useful in a variety of other vision tasks, including image deblurring, super-resolution, and anomaly detection. Inspired by the success of DPM, we propose MedSegDiff, the first DPM-based model for general medical image segmentation tasks. To enhance the step-wise regional attention in DPM for medical image segmentation, we propose Dynamic Conditional Encoding, which establishes state-adaptive conditions for each sampling step. Additionally, we propose the Feature Frequency Parser (FF-Parser) to eliminate the negative effect of high-frequency noise components in this process. We verify the effectiveness of MedSegDiff on three medical segmentation tasks with different image modalities, including optic cup segmentation over fundus images, brain tumor segmentation over MRI images, and thyroid nodule segmentation over ultrasound images. Our experimental results show that MedSegDiff outperforms state-of-the-art (SOTA) methods by a considerable performance gap, demonstrating the generalization and effectiveness of the proposed model.

**Keywords:** diffusion probabilistic model, medical image segmentation, brain tumor, optic cup, thyroid nodule

---

[*] Correspondence Author

## 1. Introduction

Medical image segmentation is the process of partitioning a medical image into meaningful regions. This is a fundamental step in many medical image analysis applications, such as diagnosis, surgical planning, and image-guided surgery. In recent years, there has been a growing interest in automatic medical image segmentation methods. These methods have the potential to reduce the time and effort required for manual segmentation and to improve the consistency, accuracy and trustworthy of results, which is significant for the black-box deep learning technology(Zhang et al., 2021, 2022a,b). With the development of deep learning techniques, more and more studies have successfully applied neural network (NN) based models to medical image segmentation tasks, from the popular convolution neural networks (CNN) (Ji et al., 2021) to the recent vision transformers (ViT) (Chen et al., 2021; Wang et al., 2021; Liu et al., 2022a; Zhao et al., 2021; Zhang et al., 2022c; Zhang et al.).

Recently, the diffusion probabilistic model (DPM) has gained popularity as a powerful class of generative models (Ho et al., 2020). These models are capable of generating images with high diversity and synthesis quality. Large diffusion models, such as DALL-E2 (Ramesh et al., 2022), Imagen (Saharia et al., 2022a), and Stable Diffusion (Rombach et al., 2022), have demonstrated incredible generation capabilities (Zhao and Shi, 2021; Goodfellow et al., 2020). Diffusion models were originally applied in fields in which there is no absolute ground truth. However, very recent studies have shown that they are also effective for problems in which the ground truth is unique, such as super-resolution (Saharia et al., 2022b), deblurring (Whang et al., 2022) , and segmentation (Amit et al., 2021).

Inspired by the recent success of DPM, we propose a unique DPM-based segmentation model for medical image segmentation tasks. To our knowledge, we are the first to propose a DPM-based model for general medical image segmentation. We note that in medical image segmentation tasks, lesions/organs are often ambiguous and difficult to discriminate from the background. In this case, an adaptive calibration process is crucial to obtain accurate results. Following this idea, we propose Dynamic Conditional Encoding over vanilla conditional DPM to design the proposed model, named MedSegDiff. In the iterative sampling process, MedSegDiff conditions each step with the image prior to learn the segmentation map. To achieve adaptive regional attention, we integrate the segmentation map of the current step into the image prior encoding at each step. Specifically, we fuse the current-step segmentation mask with the image prior on the feature level in a multi-scale manner. This allows the corrupted current-step mask to dynamically enhance the condition features and improve the reconstruction accuracy. To eliminate high-frequency noise in the corrupted mask, we propose the Feature Frequency Parser (FF-Parser) to filter the features in the Fourier space. FF-Parsers are adopted on each skip connection path for multi-scale integration.

We evaluate MedSegDiff on three medical segmentation tasks: optic cup segmentation, brain tumor segmentation, and thyroid nodule segmentation. These tasks use different modalities, including fundus images, brain MRI images, and ultrasound images, respectively. MedSegDiff outperforms the previous state-of-the-art on all three tasks with different modalities, demonstrating the generalization and effectiveness of the proposed method. In brief, the contributions of the paper are:

- We propose the first DPM-based model for general medical image segmentation.

- We introduce Dynamic Conditional Encoding strategy to enable step-wise attention.

- We propose the FF-Parser method to effectively eliminate high-frequency noise components.

- Our model achieves state-of-the-art performance on three different medical segmentation tasks with diverse image modalities.

## 2. Method

We design our model based on diffusion model proposed in (Ho et al., 2020).Diffusion models are generative models that consist of two stages: a forward diffusion stage and a reverse diffusion stage. In the forward process, a segmentation label $x_0$ is gradually corrupted by Gaussian noise through a series of $T$ steps. In the reverse process, a neural network is trained to recover the original data by reversing the noising process. This can be represented as:

$$p_\theta(x_{0:T-1}|x_T) = \prod_{t=1}^{T} p_\theta(x_{t-1}|x_t), \tag{1}$$

where $\theta$ is the set of parameters for the reverse process. Starting from a Gaussian noise distribution, $p_\theta(x_T) = \mathcal{N}(x_T; 0, I_{n \times n})$, where $I$ is the original image, the reverse process transforms the latent variable distribution $p_\theta(x_T)$ to the data distribution $p_\theta(x_0)$. To be symmetrical to the forward process, the reverse process recovers the noisy image step by step to obtain the final clear segmentation.

Following the standard implementation of DPM, we adopt a UNet as the network for the learning. An illustration is shown in Figure 1. To achieve the segmentation, we condition the step estimation function $\epsilon$ using the raw image prior, which is given by:

$$\epsilon_\theta(x_t, I, t) = D((E_t^I + E_t^x, t), t), \tag{2}$$

where $E_t^I$ is the conditional feature embedding of the raw image, $E_t^x$ is the feature embedding of the segmentation map at the current step. These two embeddings are added and passed through a UNet decoder $D$ for the reconstruction. The step index $t$ is integrated with the added embedding and decoder features, and is embedded using a shared learned look-up table, as described in (Ho et al., 2020).

### 2.1. Dynamic Conditional Encoding

In most conditional DPM, the conditional prior will be a unique feature embedding. However, medical image segmentation is known for being challenging due to ambiguous objects, with lesions or tissues often difficult to distinguish from the background. This issue is further compounded by low-contrast image modalities like MRI or ultrasound images. To address this problem, we propose a Dynamic Conditional Encoding for each step. We observe that the raw image contains accurate segmentation target information, but it is hard to distinguish from the background. Meanwhile, the current-step segmentation map contains enhanced target regions but is not entirely accurate. This motivates us to integrate

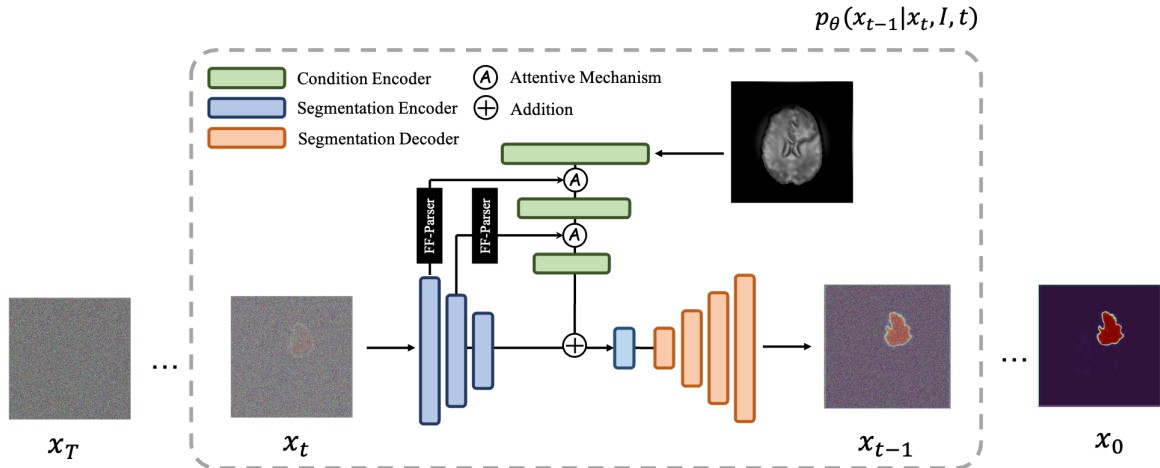

Figure 1: An illustration of MedSegDiff. For the clarity, the time step encoding is omitted in the figure.

the current-step segmentation information $x_t$ into the conditional raw image encoding for mutual complement. Specifically, we implement integration on the feature level. In the raw image encoder, we enhance its intermediate feature with the current-step encoding features. Each scale of the conditional feature map $m_I^k$ is fused with the $x_t$ encoding features $m_x^k$ of the same shape, where $k$ is the index of the layer. The fusion is achieved using an attentive-like mechanism $\mathcal{A}$. In this mechanism, we first apply layer normalization to both feature maps and then multiply them together to get an affinity map. We then multiply the affinity map with the condition encoding features to enhance the attentive region, as follows:

$$\mathcal{A}(m_I^k, m_x^k) = (LN(m_I^k) \otimes LN(m_x^k)) \otimes m_I^k, \tag{3}$$

where $\otimes$ denotes element-wise multiplication, and $LN$ denotes layer normalization. We apply this operation on the middle two stages, where each is a convolutional stage implemented following ResNet34.

This Dynamic Conditional Encoding strategy helps MedSegDiff to dynamically localize and calibrate the segmentation. But it may introduce additional high-frequency noise due to the integration of the noise-corrupted $x_t$ embedding. To address this issue, we further propose the FF-Parser to constrain the high-frequency components in the features.

### 2.2. FF-Parser

We incorporate FF-Parser into the feature integration pathway to address the issue of high-frequency noise introduced by the integration of $x_t$ embedding. FF-Parser is designed to constrain the noise-related components in the $x_t$ features. The main idea is to learn a parameterized attentive map that is applied in the Fourier-space features. Given a decoder feature map $m \in \mathbb{R}^{H \times W \times C}$, we first perform 2D FFT (fast Fourier transform) along the spatial dimensions, which can be represented as:

$$M = \mathcal{F}[m] \in \mathbb{C}^{H \times W \times C}, \tag{4}$$

where $\mathcal{F}[\cdot]$ denotes the 2D FFT. Next, we modulate the spectrum of $m$ by multiplying a parameterized attentive map $A \in \mathbb{C}^{H \times W \times C}$ to $M$:

$$M' = A \otimes M, \tag{5}$$

where $\otimes$ denotes the element-wise product. Finally, we reverse $M'$ back to the spatial domain by adopting inverse FFT:

$$m' = \mathcal{F}^{-1}[M']. \tag{6}$$

FF-Parser can be regarded as a learnable version of frequency filters, which are widely applied in digital image processing (Pitas, 2000). Unlike spatial attention, FF-Parser globally adjusts the components of specific frequencies. Thus, it can be trained to constrain the high-frequency components for adaptive integration.

### 2.3. Training and Architecture

MedSegDiff is trained following the standard process of DPM (Ho et al., 2020). Specifically, the loss can be represented as:

$$\mathcal{L} = E_{x_0, \epsilon, t}[||\epsilon - \epsilon_\theta(\sqrt{\hat{a}_t} x_0 + \sqrt{1 - \hat{a}_t}\epsilon, I_i, t)||^2]. \tag{7}$$

In each iteration, a random pair of raw image $I_i$ and segmentation label $S_i$ is sampled for training. The iteration number is sampled from a uniform distribution, and $\epsilon$ is sampled from a Gaussian distribution.

The main architecture of MedSegDiff is a modified ResUNet (Yu et al., 2019), which we implement using a ResNet encoder followed by a UNet decoder. We follow the detailed network settings of (Nichol and Dhariwal, 2021). Both $I$ and $x_t$ are encoded using two individual encoders. Each encoder consists of convolution stages containing multiple residual blocks. The number of residual blocks in each stage follows that of ResNet34. Each residual block is composed of two convolutional blocks, each one consisting of a group-normalization layer and a SiLU (Elfwing et al., 2018) activation layer, followed by a convolutional layer. The residual block receives the time embedding through a linear layer, a SiLU activation, and another linear layer. The result is then added to the output of the first convolutional block. The obtained $E^I$ and $E^{x_t}$ are added together and sent to the last encoding stage. A standard convolutional decoder is connected to predict the final result.

## 3. Experiments

### 3.1. Dataset

We conducted experiments on three different medical tasks using different image modalities: optic-cup segmentation from fundus images, brain tumor segmentation from MRI images, and thyroid nodule segmentation from ultrasound images. We evaluated the performance of our method for glaucoma, thyroid cancer, and melanoma diagnosis on the REFUGE-2 dataset (Fang et al., 2022), BraTs-2021 dataset (Baid et al., 2021), and DDTI dataset (Pedraza et al., 2015), which contain 1200, 2000, and 8046 samples, respectively. Both segmentation and diagnosis labels are publicly available in these datasets, and we split the data into train/validation/test sets following the default settings of the respective datasets.

### 3.2. Implementation Details

We experimented with four different variants of our model: MedSegDiff-L, MedSegDiff-B, and MedSegDiff-S, which correspond to huge, large, basic, and small variants, respectively. We used UNet with 4x, 5x, and 6x downsampling for MedSegDiff-S, MedSegDiff-B, and MedSegDiff-L, respectively. In our experiments, we employed 100 diffusion steps for the inference, which is much smaller than in most previous studies (Ho et al., 2020; Nichol and Dhariwal, 2021). All experiments were implemented using the PyTorch platform and trained/tested on four Tesla P40 GPUs with 24GB of memory, except for MedSegDiff-L. All images were uniformly resized to 256x256 pixels. The networks were trained end-to-end using the AdamW optimizer (Loshchilov and Hutter, 2017). MedSegDiff-B and MedSegDiff-S were trained with a batch size of 32, while MedSegDiff-L were trained with a batch size of 64. The initial learning rate was set to 1e-4. All models were run 25 times for ensemble inference, and the STAPLE algorithm (Warfield et al., 2004) was used to fuse the different samples. For fair comparison, we reproduced the diffusion-based competitor EnsemDiff (Wolleb et al., 2021) and SegDiff (Amit et al., 2021) using the same ensemble settings.

### 3.3. Main Results

We compared our approach with state-of-the-art (SOTA) segmentation methods proposed for the three specific tasks and general medical image segmentation methods. The main results are presented in Table 1. Specifically, ResUnet(Yu et al., 2019) and BEAL(Wang et al., 2019) were proposed for optic disc/cup segmentation, TransBTS(Wang et al., 2021) and EnsemDiff(Wolleb et al., 2021) were proposed for brain tumor segmentation, and MT-Seg(Gong et al., 2021) and UltraUNet(Chu et al., 2021) were proposed for thyroid nodule segmentation. CENet(Gu et al., 2019), MRNet(Ji et al., 2021), nnUNet(Isensee et al., 2021), and TransUNet(Chen et al., 2021) were proposed for general medical image segmentation. We also included SegDiff(Amit et al., 2021) and SegNet(Badrinarayanan et al., 2017), which were proposed for natural image segmentation, because of their close relationship with our method. The segmentation performance was evaluated using Dice score and IoU for 2D images, and an additional 95th percentile Hausdorff Distance (HD95) metric for 3D volumes.

In Table 1, we compare our method with those implemented with various network architectures, including CNN (ResUNet, BEAL, nnUNet, SegNet), vision transformer (TransBTS, TransUNet) and DPM (EnsemDiff, SegDiff). We can observe that advanced network architectures commonly achieve better results. For instance, in optic-cup segmentation, the ViT-based general segmentation method TransUNet performs even better than the CNN-based task-specific method BEAL. On brain tumor segmentation, the recently proposed DPM-based segmentation method EnsemDiff outperforms all previous ViT-based competitors, i.e., TransBTS and TransUNet. MedSegDiff not only adopts the successful DPM but also designs a suitable strategy specifically for the general medical image segmentation task. We can observe that MedSegDiff outperforms all other methods on the three different tasks, demonstrating its generalization capability towards different medical segmentation tasks and different image modalities. When compared to the DPM-based model proposed specifically for brain tumor segmentation, i.e., EnsemDiff, MedSegDiff improves

Table 1: The comparison of MedSegDiff with SOTA segmentation methods. Best results are denoted as **bold**. The grey background denotes the methods are proposed for that/these particular tasks.

| | Optic-Cup | | Brain-Turmor | | | Thyroid Nodule | |
| --- | --- | --- | --- | --- | --- | --- | --- |
| | Dice | IoU | Dice | IoU | HD95 | Dice | IoU |
| ResUnet | 80.1 | 72.3 | 78.4 | 71.3 | 18.71 | 78.3 | 70.7 |
| BEAL | 83.5 | 74.1 | 78.8 | 71.7 | 18.53 | 78.6 | 71.6 |
| TransBTS | 85.4 | 75.7 | 87.6 | 78.44 | 12.44 | 78.6 | 71.6 |
| EnsemDiff | 84.2 | 74.4 | 88.7 | 80.9 | 10.85 | 83.9 | 75.3 |
| MTSeg | 82.3 | 73.1 | 82.2 | 74.5 | 15.74 | 82.3 | 75.2 |
| UltraUNet | 83.1 | 73.78 | 84.5 | 76.3 | 14.03 | 84.5 | 76.2 |
| CENet | 78.6 | 69.4 | 76.2 | 68.9 | 19.35 | 78.9 | 71.2 |
| MRNet | 84.2 | 75.1 | 83.4 | 75.6 | 15.18 | 80.4 | 73.4 |
| SegNet | 80.4 | 70.7 | 80.2 | 72.9 | 17.42 | 81.7 | 74.5 |
| SegDiff | 82.5 | 71.9 | 85.7 | 77.0 | 14.31 | 81.9 | 74.8 |
| nnUNet | 84.9 | 75.1 | 88.5 | 80.6 | 11.20 | 84.2 | 76.2 |
| TransUNet | 85.6 | 75.9 | 86.6 | 79.0 | 13.74 | 83.5 | 75.1 |
| MedSegDiff-S | 81.2 | 71.7 | 82.3 | 73.6 | 15.85 | 80.8 | 73.7 |
| MedSegDiff-B | 85.9 | 76.2 | 88.9 | 81.2 | 10.41 | 84.8 | 76.4 |
| MedSegDiff-L | **86.9** | **78.5** | **89.9** | **82.3** | **8.72** | **86.1** | **79.6** |

2.3% on Dice and 2.4% on IoU, which indicates the effectiveness of our unique techniques, i.e., Dynamic Conditioning Encoding and FF-Parser.

Figure 2 showcases several typical examples generated by our proposed MedSegDiff method and other state-of-the-art (SOTA) methods. As can be observed, the target lesions and tissues in these medical images are often ambiguous and difficult to identify even for human eyes. However, our proposed method outperforms other computer-aided methods, generating more accurate segmentation maps, especially in the ambiguous regions. This improvement can be attributed to the effective combination of Dynamic Conditional Encoding and FF-Parser with DPM, which allows for better localization and calibration of the segmentation on low-contrast or ambiguous images.

### 3.4. Ablation Study

We conducted a comprehensive ablation study to verify the effectiveness of the proposed Dynamic Conditional Encoding and FF-Parser. We used a basic conditional-diffusion model proposed in SegDiff (Amit et al., 2021) as the baseline and incrementally added the proposed modules over it. An illustration is shown in Fig. 3. We evaluated the performance using Dice score (%) on all three tasks, and the quantitative comparison results are shown in Table 2. We use Dy-Cond to denote Dynamic Conditional Encoding. From the table, we can see that Dy-Cond provides considerable improvements over baseline. For tasks where region localization is important, such as optic-cup segmentation, it improves the performance by 1.9%. For tasks with low-contrast images, such as brain tumor and thyroid

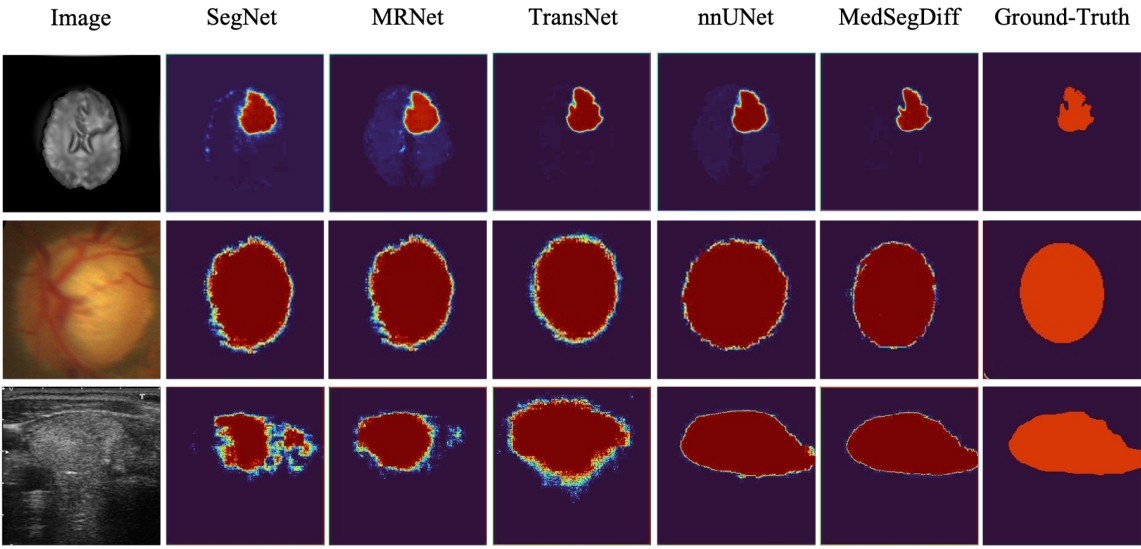

Figure 2: The visual comparison of competing general medical image segmentation methods in Table 1. From top to down are brain-tumor segmentation, optic-cup segmentation and thyroid nodule segmentation, respectively.

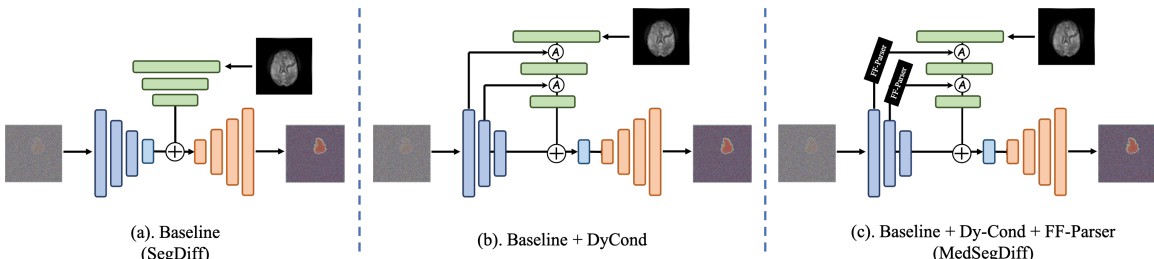

Figure 3: An illustration of the ablation study. Dy-Cond denotes the proposed Dynamic Conditional Encoding strategy.

nodule segmentation, Dy-Cond improves the performance by 1.8% and 1.7%, respectively. These results indicate that Dy-Cond is a generally effective strategy for DPM in both cases. FF-Parser, which is built on top of Dy-Cond, mitigates high-frequency noise and further optimizes the segmentation results. It helps MedSegDiff achieve the best performance on all three tasks, with an additional improvement of nearly 1%.

Table 2: An ablation study on Dynamic Condition Encoding and FF-Parser. Dice score(%) is used as the metric.

| Name | Dy-Cond | FF-Parser | OpticCup | BrainTumor | ThyroidNodule |
|---|---|---|---|---|---|
| (a). SegDiff | | | 82.5 | 85.7 | 81.9 |
| (b). | ✓ | | 84.4 | 87.5 | 83.6 |
| (c). MedSegDiff (proposed) | ✓ | ✓ | **85.9** | **88.9** | **84.8** |

## 4. Conclusion

In this paper, we present MedSegDiff, a scheme for DPM-based general medical image segmentation, that incorporates two novel techniques: Dynamic Conditional Encoding and FF-Parser, to improve segmentation performance. Our comparison experiments on three medical image segmentation tasks with different image modalities demonstrate that our model outperforms previous SOTA methods. Being the first DPM-based application in general medical image segmentation, we believe that MedSegDiff will serve as an essential benchmark for future research in this field.

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

## 5. Appendix

### 5.1. Detailed Analysis and Discussion

#### 5.1.1. The inference time and model complexity

Diffusion model requires more inference time than a traditional neural network due to its iterative nature. In Table 3, we provide a comparison of model complexity and inference time with competing segmentation methods. The inference time is recorded for processing a single $256 \times 256$ image or slice of a volume. Despite not increasing the model complexity, MedSegDiff requires much more inference time, which is a natural limitation of all diffusion-based methods and remains a pressing challenge in the research community.

Table 3: Comparison of parameters and inference times with competing segmentation methods. The unit of parameter numbers is million. The unite of inference time is seconds per image.

| Model | Params (M) | Inference Time (s/Image) |
|---|---|---|
| ResUNet | 12 | 0.02 |
| MRNet | 48 | 0.09 |
| nnUNet | 19 | 0.03 |
| TransUNet | 96 | 0.08 |
| MedSegDiff-S | 13 | 12.50 |
| MedSegDiff-B | 25 | 16.27 |
| MedSegDiff-L | 41 | 19.76 |

We have also attempted to apply several recently proposed DPM accelerating algorithms, including DPM-Solver (Lu et al., 2022), Flow-S&F (Liu et al., 2022b), and DDSS (Watson et al., 2022), to MedSegDiff-B, as shown in Table 4. However, we found that in our conditional-DPM segmentation setting, these algorithms did not perform as well as they did in the image generation case. This suggests that the Conditional-DPM variants may require specific acceleration algorithms. We consider this to be a very interesting finding and plan to explore it further in our future work.

Table 4: The performance is evaluated by Dice Score (%) on three datasets, with the value in [ ] indicating the reduction in performance for faster inference.

| Model | Optic-Cup | Brain-Tumor | Thyroid Nodule | Inference Time (s/Image) |
|---|---|---|---|---|
| DPM-solver (Neurips 2022) | 83.69 [-2.26] | 84.71 [-4.26] | 81.95 [-2.88] | 6.30 |
| Flow-S&F (ICLR2023) | 84.13 [-1.82] | 85.49 [-3.48] | 82.27 [-2.56] | 8.67 |
| DDSS (ICLR2022) | 84.63 [-1.32] | 85.27 [-3.70] | 82.06 [-2.77] | 5.81 |

#### 5.1.2. Effect of ensemble

Similar to the previous observation(Wolleb et al., 2021), we find ensemble of different runs plays an important role in diffusion-based prediction. However, it is important to note that

while both are referred to as ensemble, there are significant differences between diffusion ensemble and traditional model ensemble. Diffusion model ensemble fuses predictions from multiple runs of a single model, while traditional ensemble fuses predictions from multiple different models.

We quantitatively verify the effect of ensemble over MedSegDiff-L in Table 5. We run each setting 20 times to compute its variance. We find the model performance increases rapidly in the first 5 ensemble (commonly comparative to SOTA) and then stabilizes. The best performance is commonly achieved after about 25 ensemble. Besides the performance improvement, the increased ensemble size also leads to smaller performance variance. For instance, the variance of BraTs decreases from 0.74% for a single sample to 0.52% for a 5-run ensemble and further drops to 0.40% for a 25-run ensemble. Over 25 runs, more times of ensemble can slightly mitigate the variance but cannot improve the mean performance anymore. Another noteworthy finding is that STAPLE algorithm(Warfield et al., 2004), which previously used for multi-rater fusion, can effectively improve performance and mitigate variance when used for ensembles. For example, STAPLE improves the 0.53% mean performance and mitigates 0.6% variance for 5-run ensemble over BraTs. The effectiveness of STAPLE will decrease as the number of ensemble increases, which is intuitive as more ensembles naturally lead to better consensus.

Table 5: The comparison of different ensemble settings. The performance is evaluated over three datasets by mean Dice Score (%) $\pm$ its variance.

| Ensemble | Optic-Cup | Brain-Tumor | Thyroid Nodule |
|---|---|---|---|
| one sample | $84.65 \pm 1.04$ | $87.65 \pm 0.74$ | $84.45 \pm 0.69$ |
| 5 runs | $85.78 \pm 0.56$ | $88.60 \pm 0.52$ | $85.27 \pm 0.41$ |
| 5 runs + STAPLE | $86.61 \pm 0.44$ | $89.13 \pm 0.46$ | $85.84 \pm 0.27$ |
| 25 runs | $86.42 \pm 0.29$ | $89.76 \pm 0.40$ | $85.95 \pm 0.21$ |
| 25 runs + STAPLE | $\mathbf{86.94} \pm 0.25$ | $\mathbf{89.92} \pm 0.37$ | $86.18 \pm 0.16$ |
| 50 runs | $86.80 \pm 0.22$ | $89.90 \pm 0.33$ | $86.16 \pm 0.15$ |
| 50 runs + STAPLE | $86.84 \pm \mathbf{0.21}$ | $89.91 \pm \mathbf{0.31}$ | $\mathbf{86.20} \pm \mathbf{0.12}$ |

### 5.2. Implement Details

We provide the implement details of the comparing methods:

- ResUNet: During model training, we used Adam optimization with an initial learning rate of 0.05. we set the training batch size to 32. The backbone of the framework is ResNet-34 with a standard UNet decoder. We employ the two-steps training strategy following their paper. We train its Step 1 and Step 2 with 50 and 100 epochs on REFUGE-2 dataset, 70 and 140 epochs on BraTs dataset, and 60 and 110 epochs on DDIT dataset, respectively.

- BEAL: We trained the framework using a minibatch size of 32. The discriminators De and Db were optimized using the SGD algorithm, while the segmentation network was optimized using the Adam optimizer. The initial learning rate for SGD was set

to 1e-3 and decreased by a factor of 0.2 every a half of the epochs for a total of 100, 140, and 110 epochs for REFUGE-2, BraTs2021, and DDIT datasets, respectively. The learning rate for discriminator training was set to 2.5e-5.

- TransBTS: We trained the model using a batch size of 32 and the Adam optimizer. We set the initial learning rate to $10^{-4}$ and used a polynomial learning rate strategy, with the initial rate decaying by each iteration with power 0.9. The model is trained with 2000, 8000, and 2000 epochs on REFUGE-2, BraTs2021, and DDIT datasets, respectively.

- EnsemDiff: We used a linear noise schedule for T = 1000 steps. The model is trained using the hybrid loss objective, with a learning rate of $10^{-4}$ for the Adam optimizer, and a batch size of 32. The first layer has 128 channels, and we use one attention head at a resolution of 16. We train the model for 60,000 iterations on all three datasets.

- MTSeg: As suggested by the authors, the model is initialized using Xavier initialization. We optimized the model using SGD with a learning rate of 0.001 and a batch size of 32. The model is trained for a total of 100, 140 and 110 epochs for REFUGE-2, BraTs2021, and DDIT datasets, respectively.

- UltraUNet: The backbone of the model is a standard UNet with 4 down-sampling modules. We train the model using a batch size of 32 with Adam optimizer. The initial learning rate is set as $10^{-4}$. The model is trained for a total of 100, 140 and 110 epochs for REFUGE-2, BraTs2021, and DDIT datasets, respectively.

- CE-Net: We utilized mini-batch stochastic gradient descent (SGD) with a batch size of 32, momentum of 0.9, and weight decay of 0.0001. We employed the poly learning rate policy, where the learning rate is multiplied by (1 - iter/max_iter)p̂ower with a power of 0.9 and an initial learning rate of 4e-3. The maximum number of epochs was set to 100, 140, and 110 epochs for REFUGE-2, BraTs2021, and DDIT datasets, respectively.

- MRNet: The framework utilizes the U-Net architecture with ResNet34 as the backbone, and the MRM module adopts the DeepLab-V3+ architecture with VGG-16 as the backbone. The network is trained in an end-to-end manner using the Adam optimizer. For training, we set a mini-batch size of 32 and use 100, 140, and 110 epochs for REFUGE-2, BraTs2021, and DDIT datasets, respectively. The learning rate is initially set as $10^{-4}$.

- SegNet: We train the SegNet as proposed in (Kendall et al., 2015), with a learning rate of $10^{-4}$ for the Adam optimizer and a batch size of 32. Training is performed with the binary cross-entropy loss and is stopped after 100, 140, and 110 epochs for REFUGE-2, BraTs2021, and DDIT datasets, respectively.

- SegDiff: We trained SegDiff using the AdamW optimizer with a learning rate of $10^{-4}$, and the same backbone setting as MedSegDiff for a fair comparison. The batch size was set to 32, unless specified otherwise, and 64 for SegDiff++. The model was trained for 60,000 iterations on all three datasets.

- nnU-Net: We take over all hyper-parameter settings as proposed in their official implementation, which can be found at https://github.com/MIC-DKFZ/nnUNet. The configurations are automatically set by their pipeline.

- TransUNet: We use a hybrid encoder design with combined ResNet-50 and ViT. The models are trained using the SGD optimizer with a learning rate of 0.01, momentum of 0.9, and weight decay of 1e-4. The batch size is 32, and the number of training epochs are 100, 140, and 110 for REFUGE-2, BraTs2021, and DDIT datasets, respectively.

