# OpenReview forum: "MedSegDiff: Medical Image Segmentation with Diffusion Probabilistic Model"
_MIDL.io/2023/Conference — MIDL 2023 Poster_

### Official Review · Reviewer_YZZy · 2023-02-03

**Confidence:** 3
**Preliminary Rating:** 2

**Summary:**

This paper proposes the use of a diffusion probabilistic model (DPM) for the segmentation of medical images. The paper claims to be the first using DPMs for the task. The proposed framework introduces a novel way to condition the model on the input image and a Fourier  transform-based module  that allows to denoise the image by removing high frequency components.

The method is evaluated on three well-known medical imaging datasets reporting state of the art results.

**Strengths:**

* The work brings the use of diffusion models to medical image segmentation, and more generally to segmentation itself.
* Good set of works used for comparison in the benchmark
* tested on multiple datasets

**Weaknesses:**

* The paper is not well written and it contains numerous typos. This makes the paper difficult to understand.
* The paper shares many of the concepts presented in [1], in particular the use of DPMs for segmentation and the conditioning of the model. Positioning w.r.t this work and what is original in the current one should be clearly stated

[1] Amit, Tomer, et al. "Segdiff: Image segmentation with diffusion probabilistic models." arXiv preprint arXiv:2112.00390 (2021)

**Deanonymize Review:**

no

**Detailed Comments:**

- Figure 2 could be removed. It does not add any much information
- Ablation models are not well explained. It is difficult to grasp the difference between each of them
- Although the benchmark is quite complete, it would be good to include the best performing methods of each image segmentation task in the comparison.

**Justification of score**
I appreciate the effort made by the authors to address the points raised during the review. However, one of my main concerns, which is I consider that the paper holds many similarities with [1] (see below), has not been discussed during the rebuttal. I think this is a critical point, thus motivating the final score.

[1] Amit, Tomer, et al. "Segdiff: Image segmentation with diffusion probabilistic models." arXiv preprint arXiv:2112.00390 (2021)

**Paper Type:**

methodological development

**Questions To Address In The Rebuttal:**

1) What are your contributions and novelty of your work with respect to [1]?


[1] Amit, Tomer, et al. "Segdiff: Image segmentation with diffusion probabilistic models." arXiv preprint arXiv:2112.00390 (2021)

---

### Official Review · Reviewer_qM3c · 2023-02-03

**Confidence:** 4
**Preliminary Rating:** 5
**Recommendation:** Poster

**Summary:**

The authors propose a novel diffusion probabilistic model for Medical Image Segmentation
(MedSeg-Diff). It combines the DPM-based architecture with a conditional encoding strategy for
step-wise attention
and an FF-Parser to eliminate the negative effects of high-frequency components. They show the
effectiveness of the proposed method MedSeg-Diff by comparing it to 11 other SOTA methods on
three other datasets, where it outperforms them in all cases in terms of Dice accuracy and IoU.

**Strengths:**


- Very thorough comparison to 11 other state-of-the-art methods
- Clear motivation why dynamic conditioning and FF-Parser have been introduced
The ablation study shows the effectiveness of the proposed dynamic conditioning and FFParser, which together improve Dice Accuracy on both datasets by 2%
- The proposed method shows promising results on three modalities of medical imaging
segmentation datasets and outperforms the other state-of-the-art methods evaluated.

**Weaknesses:**

- While the evaluation is extensive, the authors do not explain why only one dataset is
evaluated for ResUnet, BEAL, TransBTS, EnsemDiff, MTSeg and UltraUNet. Especially since
EnsemDiff gets close to the MedSeg performance.
- The evaluation setup is not very clear for many methods, for example the authors don't
mention which setup of the nnUNet has been evaluated, and this is the case for most other
non Diff methods.
- Writing should be improved
- Many spelling errors are present

**Deanonymize Review:**

no

**Paper Type:**

methodological development

**Questions To Address In The Rebuttal:**

- The setup of the other SOTA methods should be defined more clearly; also, an explanation
should be given as, why some are only run on one of the three datasets.
- The spelling error should be corrected and in general, parts of the text should be rewritten.

The authors succeeded in solving all the major problems I had with their submission. They provided the missing results for all six baselines. They added a detailed description of the setups of the different baselines evaluated, and finally they had their text revised by a native speaker.
Great work.

---

### Official Review · Reviewer_hnvi · 2023-02-03

**Confidence:** 4
**Preliminary Rating:** 4
**Recommendation:** Poster

**Summary:**

In this paper, the authors propose a diffusion probabilistic model (DPM) for medical image segmentation task, named MedSegDiff. Two novel components are presented in the MedSegDiff: (1) a dynamic condition encoding for enhancing regional attention in DPM and (2) a feature frequency parser for eliminating the negative effect of high-frequency noise component. Experiments on three datasets show that the proposed method achieves state-of-the-art performance compared to existing methods. The authors claim that this is the first work that uses DPM for medical segmentation tasks.

**Strengths:**

- The method is interesting and indicates great potential of diffusion models for medical segmentation tasks.
- The experimental results show that the proposed method achieves state-of-the-art performance on benchmarks on three datasets.

**Weaknesses:**

- The benefits of the dynamic conditional encoding and feature frequency parser are hard to justify (as shown in the ablation study). The performance gain can be obtained by having a larger number of parameters instead of the mechanism as stated by the authors.

- The number of model parameters and inference time should be compared and discussed. As stated in the paper, during inference time, 25 models are used for model ensemble and each one requires 100 diffusion steps for inference. This can even take more inference time for 3D medical images as the proposed method is 2D.

- Only region-based evaluation metrics are used: Dice and IoU. A distance-based metric such as HD and ASSD should be used to understand the segmentation performance better.

**Deanonymize Review:**

no

**Detailed Comments:**

- It would be very interesting to have a comparison of the inference time and model complexity.

- The impact of model ensembles to the segmentation performance needs to be further clarified. It seems that the model ensemble was only applied to the diffusion competitor EnsemDiff but not other competing methods. It is interesting to know the performance with and without model ensemble and STAPLE.

**Paper Type:**

methodological development

**Questions To Address In The Rebuttal:**

- As mentioned in the detailed comments, please discuss the inference time and model complexity, and please clarify the impacts of model ensemble.
- It would be better to include a distance-based evaluation metric.

---

### Meta-Review · Area_Chair_9jnM · 2023-02-25

**Recommendation:** Accept (Poster)
**Confidence:** 4

**Metareview:**

Overall, the reviewers agree that this is an interesting method, which includes new technical components and points to the potential of diffusion models for medical image segmentation tasks. I concur with this. The experiments and comparisons are quite comprehensive, and include sensible ablations. Also, the authors provided comprehensive and satisfying answers during the discussion, including additional results and comparisons.

In regard to the critical point brought by Reviewer YZZy (similarity with SegDiff [1]): The authors provided a satisfying explanation of the differences between the proposed method and SegDiff. Moreover, they provided a comparison in Table 2 and an illustration in Figure 3 to further emphasize these differences. Therefore, the similarity with SegDiff should not be a ground for rejection.

Even if the technical novelty might be perceived as limited in comparison to SegDiff (which was developed for natural images), the experimental results in the context of medical imaging are, in my opinion, worth sharing with the MIDL community.